# Molecular Epidemiology of Plasmid-Mediated Types 1 and 3 Fimbriae Associated with Biofilm Formation in Multidrug Resistant *Escherichia coli* from Diseased Food Animals in Guangdong, China

Wen-Ying Guo,[a,d] Hui Zhang,[a,d] Ming Cheng,[a,d] Min-Rui Huang,[a,d] Qian Li,[a,d] Yu-Wei Jiang,[a,d] Ji-Xing Zhang,[a,d] Ruan-Yang Sun,[a,d] Min-Ge Wang,[a,d] Xiao-Ping Liao,[a,b,d] Ya-Hong Liu,[a,b,c,d] Jian Sun,[a,b,c,d] Liang-Xing Fang[a,b,c,d]

[a]National Risk Assessment Laboratory for Antimicrobial Resistance of Animal Original Bacteria, South China Agricultural University, Guangzhou, China
[b]Guangdong Laboratory for Lingnan Modern Agriculture, Guangzhou, China
[c]Jiangsu Co-Innovation Center for the Prevention and Control of Important Animal Infectious Diseases and Zoonoses, Yangzhou University, Yangzhou, China
[d]Guangdong Provincial Key Laboratory of Veterinary Pharmaceutics Development and Safety Evaluation, South China Agricultural University, Guangzhou, China

Wen-Ying Guo and Hui Zhang contributed equally to this work. Author order was determined alphabetically.

**ABSTRACT** Types 1 and 3 fimbriae in Enterobacteriaceae play versatile roles in bacterial physiology including attachment, invasion, cell motility as well as with biofilm formation and urinary tract infections. Herein, we investigated the prevalence and transmission of plasmid-mediated types 1 and 3 fimbriae from 1753 non-duplicate Enterobacteriaceae from diseased food Animals. We identified 123 (7.01%) strong biofilm producers and all was identified as *E. coli*. WGS analysis of 43 selected strong biofilm producers revealed that they harbored multiple ARGs, including ESBLs, PMQR and *mcr-1*. The gene clusters *mrkABCDF* and *fimACDH* encoding types 1 and 3 fimbriae, respectively, were identified among 43 (34.96%) and 7 (5.7%) of 123 strong biofilm isolates, respectively. These two operons were able to confer strong biofilm-forming ability to an *E. coli* weak-biofilm forming laboratory strain. Plasmid analysis revealed that *mrk* and *fim* operons were found to co-exist with ARGs and were primarily located on IncX1 and IncFII plasmids with similar backbones, respectively. *mrkABCDF* operons was present in all of 9457 *Klebsiella pneumoniae* using archived WGS data, and shared high homology to those on plasmids of 8 replicon types and chromosomes from 6 Enterobacteriaceae species from various origins and countries. In contrast, *fimACDH* operons was present in most of *Enterobacter cloacae* (62.15%), and shared high homology to those with only a small group of plasmids and Enterobacteriaceae species. This is the first comprehensive report of the prevalence, transmission and homology of plasmid-encoded type 1 and 3 fimbriae among the Enterobacteriaceae. Our findings indicated that plasmid-encoded *mrkABCDF* and *fimACDH* were major contributors to enhanced biofilm formation among *E. coli* and these two operons, in particular *mrk* could be as a potential anti-biofilm target.

**IMPORTANCE** Biofilms allow bacteria to tolerate disinfectants and antimicrobials, as well as mammalian host defenses, and are therefore difficult to treat clinically. Most research concerning biofilm-related infections is typically focused on chromosomal biofilm-associated factors, including types 1 and 3 fimbriae of biofilm-forming Enterobacterium. However, the transmission and homology of the mobile types 1 and 3 fimbriae among Enterobacteriaceae is largely unknown. The findings revealed that the plasmid-encoded type 3 fimbriae encoded by *mrkABCDF* and type 1 fimbriae encoded by *fimACDH* were major contributors to enhancing biofilm formation among strong biofilm *E. coli* from diseased food producing animals. Additionally, *mrk* operon with high homology at an amino acid sequence was present both on plasmids of various replicon types and on chromosomes

Address correspondence to Jian Sun, jiansun@scau.edu.cn, or Liang-Xing Fang, fanglx@scau.edu.cn.

The authors declare no conflict of interest.

from diverse Enterobacteriaceae species from numerous origins and countries. These findings provide important information on the transmission of the mobile types 1 and 3 fimbriae among Enterobacteriaceae, indicating a potential antibiofilm target.

**KEYWORDS** biofilm, Type 1 fimbriae, Type 3 fimbriae, plasmid-mediated, *Escherichia coli*, food animal

Biofilms are structured consortiums of embedded bacteria that are a survival strategy for many bacterial and fungal species, and are an adaptive response to constantly changing and potentially hostile environments (1). The biofilm lifestyle allows the bacteria to tolerate disinfectants and antimicrobials, as well as mammalian host defenses, and are therefore difficult to treat clinically (2, 3). Biofilms that form within the host have been implicated in serious and persistent infectious diseases including urinary tract infections (UTI), cystic fibrosis, and endocarditis (4). In the environment, biofilms can serve as reservoirs for pathogens and can contaminate surfaces and the water environment (5). Furthermore, biofilms are also involved in food contamination of fresh fruits and vegetables as well as animal-derived food products (4, 6, 7). Hence, it is critically important to explore the potential molecular mechanisms associated with biofilm formation and to design or screen antibiofilm molecules with the goal of minimizing and eradicating biofilm-related infections.

Most current research concerning biofilm-related infections is typically focused on chromosomal biofilm-associated factors of biofilm-forming bacteria (8, 9). Interestingly, several studies have also reported that conjugative plasmids can promote biofilm formation on abiotic substance through the formation of surface fimbriae (10–12). Pili or fimbriae play versatile roles in bacterial physiology, and these can be associated with attachment, invasion, and cell motility, as well as with biofilm formation (13, 14). In particular, the gene clusters that encode types 1 and 3 fimbriae in Enterobacteriaceae members are not only chromosomally encoded but also on plasmids. The first plasmid-borne type 1 gene cluster was found on an IncFII plasmid (pE110019_66) from an atypical Enteropathogenic *Escherichia coli* isolate (15). Subsequently, our laboratory identified a similar plasmid-borne type 1 fimbriae cluster *fimACDH* that could promote biofilm formation and was confirmed using the novel EZ-Tn5 transposon technique (11).

There is currently a paucity of information regarding the prevalence and distribution of plasmid-encoded type 1 fimbrial gene in the Enterobacteriaceae. However, the first plasmid-borne type 3 fimbriae encoded by the *mrkABCDF* operon was identified as the IncX1 plasmid pOLA52 from a pig *E. coli* isolate in Denmark (10, 16). This operon was found to enhance biofilm formation and likely was mobilized by a composite transposon Tn*6011*, from the chromosome of *Klebsiella pneumoniae* onto the IncX1 plasmid. The *mrkABCDF* operon has since been found in the chromosome for other Enterobacteriaceae as well as on conjugative plasmids (17, 18).

These data have indicated that type 1/3 fimbriae-encoding gene clusters can be located on conjugative plasmids and this could accelerate the spread of these two operons among Enterobacteriaceae members. In this present study, we investigated the biofilm forming abilities and drug susceptibilities in a large collection of Enterobacteriaceae isolates from diseased food producing animals. We further explored the prevalence, function, evolution, and transmission of plasmid-encoded type 1/3 fimbriae.

## RESULTS

**Biofilm formation and antimicrobial susceptibility.** We examined a total of 1,753 Enterobacteriaceae isolates for biofilm formation in Luria Bertani (LB) medium using crystal violet staining. We found that 123 (7.02%) were strong-biofilm forming ($OD_{590} \geq 0.38$) strains, and all these were identified as *E. coli*. There were 100 (5.7%) isolates that were considered moderate-biofilm formers ($0.19 \leq OD_{590} < 0.38$), and the remaining 1,530 isolates (87.28%) were weak or non-biofilm producers ($OD_{590} \leq 0.095$). The strong biofilm producers were grouped across three time periods (2002–2010, 2010–2018, 2019) with a prevalence of 7.19% (41/570), 6.77% (31/458), and 7.02% (51/725), respectively (Fig. 1a). The strong

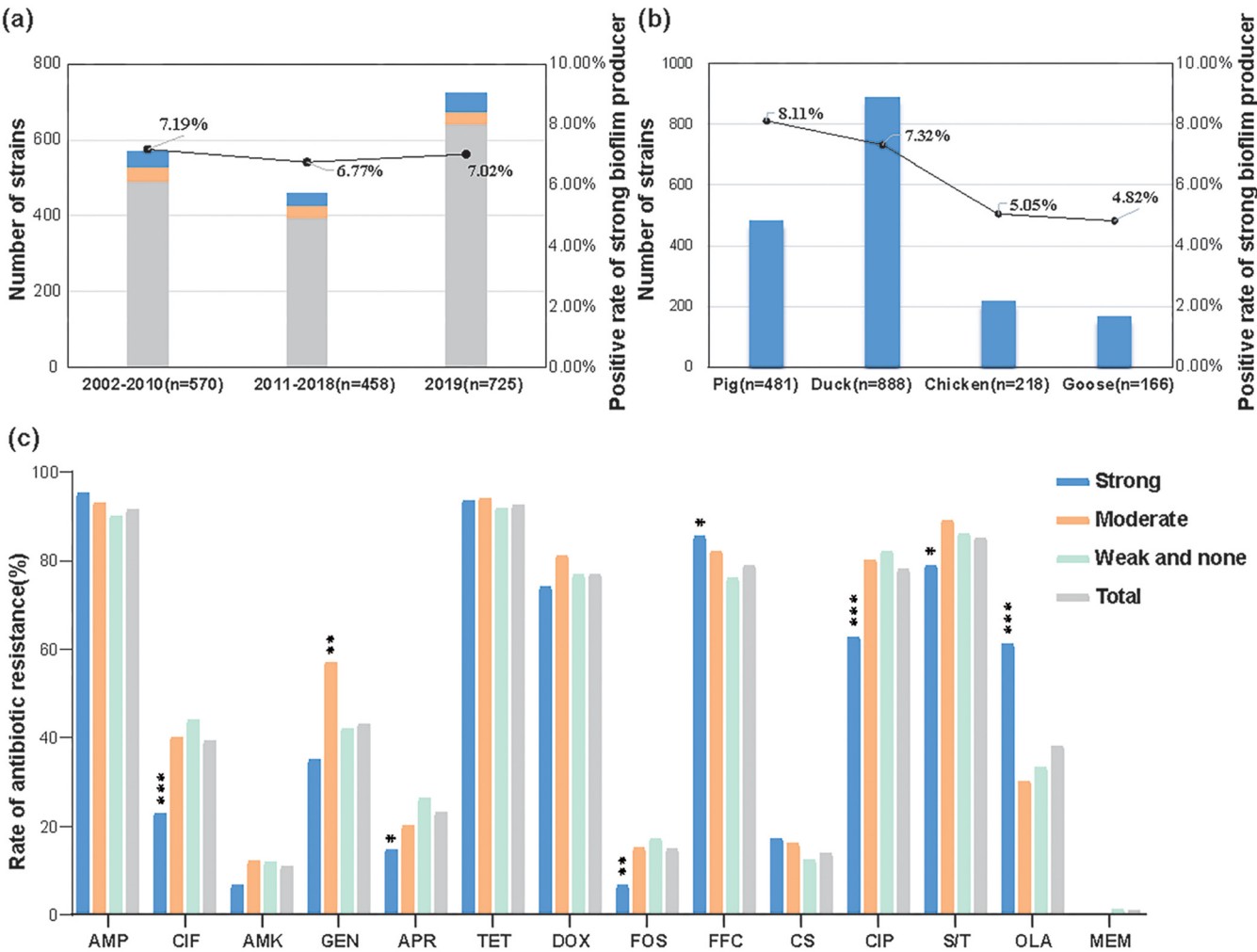

**FIG 1** Prevalence of strong biofilm producers and antibiotic resistance among Enterobacteriaceae strains from food-producing animals in the period 2002–2019. (a) Prevalence of strong biofilm producers grouped according to isolation time. (b) Prevalence of strong biofilm producers from different sources. (c) Antibiotic resistance among Enterobacteriaceae strains possessed different biofilm formation abilities. AMP, ampicillin; CIF, ceftiofur; AMK, amikacin; GEN, gentamicin; APR, apramycin; TET, tetracycline; DOX, doxycycline; FFL, florfenicol; CIP, ciprofloxacin; OLA, olaquindox; S/T, sulfamethoxazole-trimethoprim; FOS, fosfomycin; CST, colistin; MEM, meropenem. *, $P < 0.05$; **, $P < 0.01$; ***, $P < 0.001$. $P$ values were determined using the $\chi^2$ test.

biofilm-formers grouped according to source were 8.11% (39/481) in pigs, 7.32% (65/888) in ducks, 5.05% (11/218) in chickens, and 4.82% (8/166) in geese (Fig. 1b).

The 123 strong and 100 moderate biofilm-forming isolates and 408 randomly selected weak and none biofilm producers were selected for antimicrobial susceptibilities. Most of the tested isolates were resistant to ampicillin, tetracycline, florfenicol, sulfamethoxazole-trimethoprim, doxycycline, and ciprofloxacin (>70%). In contrast, resistance was lower to gentamicin (42.95%), ceftiofur (39.14%), olaquindox (38.03%), apramycin (22.98%), colistin (13.79%), amikacin (10.78%), fosfomycin (14.74%), and meropenem (0.79%). Interestingly, the group of strong biofilm producers had significantly higher prevalence of resistance to florfenicol and olaquindox and a lower prevalence of resistance to ceftiofur, ciprofloxacin, fosfomycin, apramycin, and sulfamethoxazole-trimethoprim when compared to the other isolates ($P < 0.05$) (Fig. 1c).

**WGS analysis of strongly biofilm-forming *E. coli* isolates.** A total of 42 strong biofilm-forming *E. coli* isolates were selectively sequenced. Whole-genome sequencing (WGS) analysis revealed that *mrkABCDF* (5.45 kb) and *fimACDH* (4.93 kb) were identified among 15 (28.57%) and 2 (4.76%) isolates, respectively. Additionally, two isolates harbored incomplete type 3 fimbriae operons. This group of 42 strong biofilm producers possessed 29 distinct antibiotic resistance genes (ARGs), and most carried multiple ARGs including the clinically relevant *mcr-1*

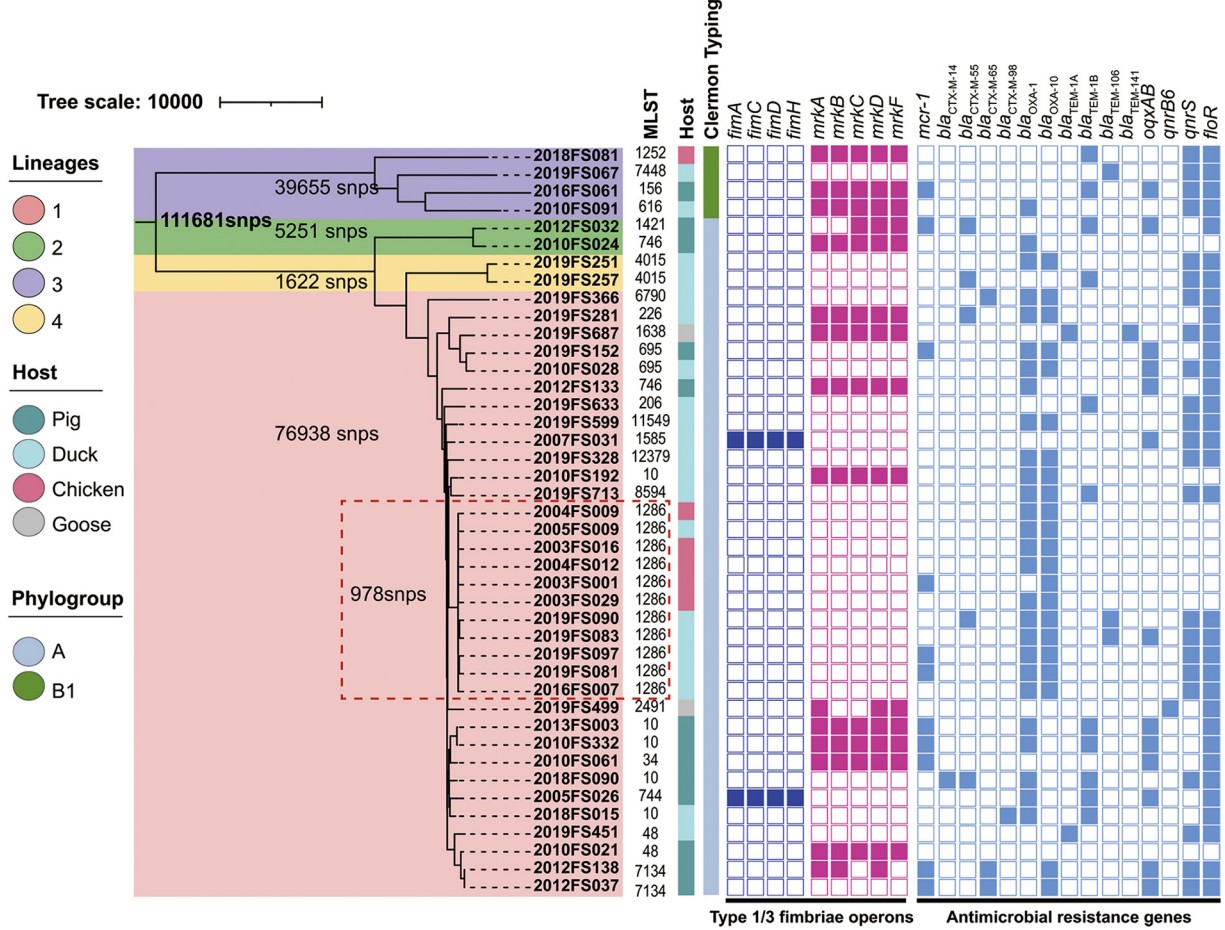

**FIG 2** Population structure and the presence of *fimACDH* and *mrkABCDF* and ARGs for the *E. coli* strong biofilm formers identified in this study.

(n = 11), *bla*CTX-M-9G/1G (n = 8), *oqxAB* (n = 13), *qnrS* (n = 23), *qnrB6* (n = 1), and *floR* (n = 33). In addition, *bla*CTX-M-55 (n = 2), *oqxAB* (n = 7), *qnrS* (n = 5), *qnrB6* (n = 1), and *floR* (n = 12) were also present among a subgroup of 15 *mrk*-positive isolates. Two *fim* isolates possessed both *oqxAB* and *floR* (Fig. 2).

Multilocus sequence typing (MLST) analysis revealed 24 sequence types (STs) including a new ST (ST12379) among the 42 strong biofilm-forming *E. coli* isolates, and ST1286 was the most prevalent (11, 26.19%), followed by ST10 (5, 11.90%). These 42 *E. coli* isolates were classified into two phylogenetic groups with the majority belonging to the commensal phylogenetic groups A (38/42, 90.48%) and B1 (4/42, 9.52%). We further analyzed population structures by constructing phylogenetic trees based on the core genomes of these 42 strong biofilm-forming *E. coli* isolates. Bayesian analysis revealed four distinct lineages, and the major lineage (lineage I) contained 34 isolates that included 11 ST1286 *E. coli* isolates that were distributed across 5 different sampling times and that possessed extremely high genetic similarity (SNPs ≤ 978). The 15 *mrk* and 2 *fim* isolates comprised 14 different STs and were distributed among 3 lineages indicating a high degree of WGS heterogeneity (Fig. 2).

**Function and prevalence of type 1/3 fimbriae operons.** To determine whether the *mrk* and *fim* operons were direct contributors to strong biofilm-forming ability, we cloned the *mrkABCDF* and *fimACDH* gene clusters from 2 strong *E. coli* biofilm forming strains that were then introduced onto a plasmid vector into the weak biofilm forming *E. coli* strain DH5α. Interestingly, both the *mrk* and *fim* gene cassettes converted DH5α to a strong biofilm former (Fig. 3a and b). Scanning electron microscopy (SEM) photomicrographs confirmed that the enhanced biofilm growth of transformants harboring *mrkABCDF* and *fimACDH* correlated

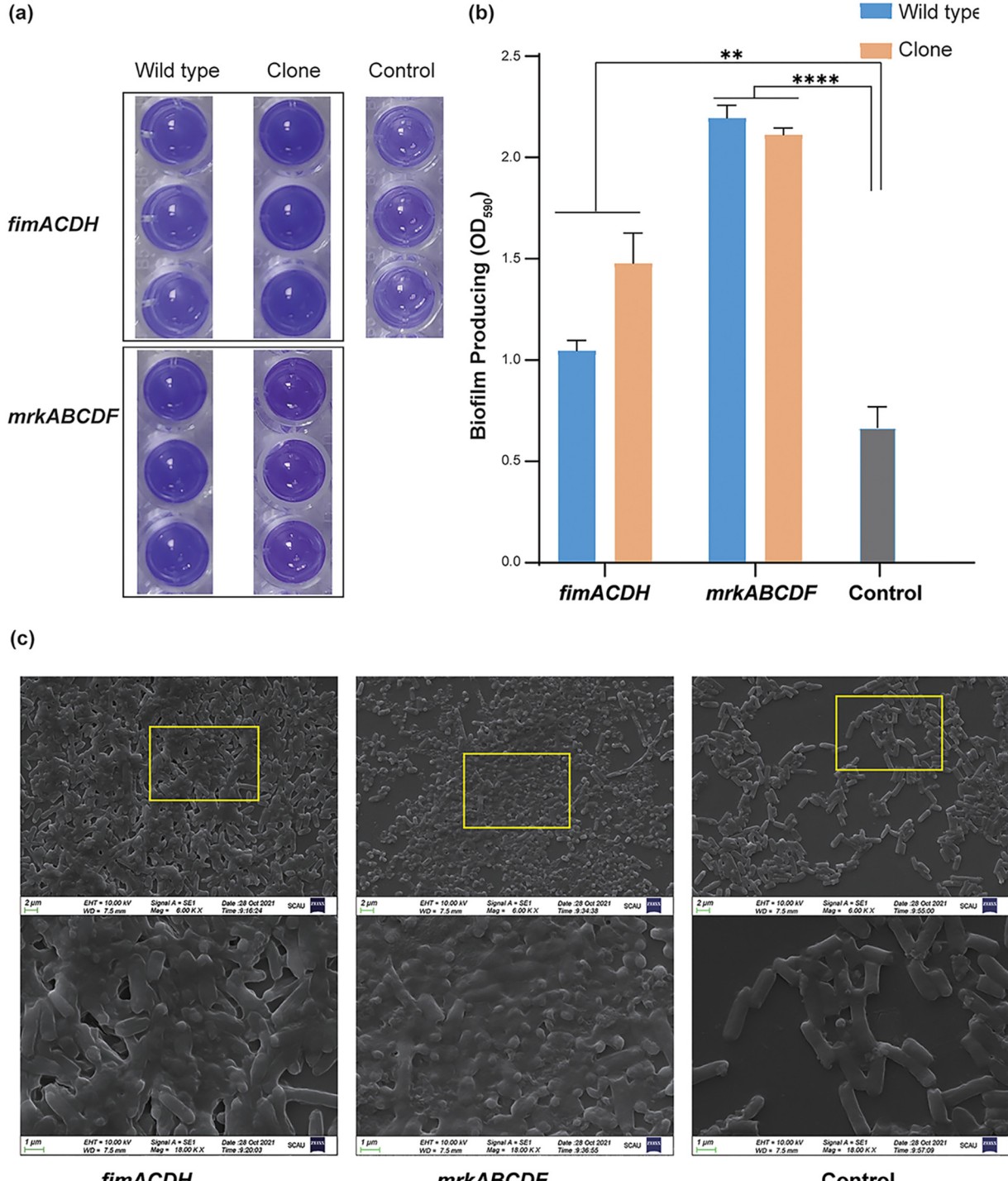

**FIG 3** Biofilm formation for cloned copies of *mrkABCDF* and *fimACDH* in the laboratory *E. coli* strain DH5$\alpha$. (a) Representative examples of crystal violet-stained biofilms and (b) quantification using absorbance at $OD_{590}$ nm. (c) Scanning electron photomicrographs of representative biofilms from *E. coli* DH5$\alpha$ containing cloned plasmid copes of *fimACDH*, *mrkABCDF*, and the PMD19-empty vector control as indicated. **, $P < 0.01$; ****, $P < 0.0001$ tested by one-way analysis of variance.

with a more densely packed arrangement of cells compared with the empty vector control (Fig. 3c).

In our population of 123 *E. coli* that were strong biofilm producers, we were able to identify complete *mrkABCDF* and *fimACDH* clusters in 43 (34.96%) and 7 (5.7%) isolates, respectively. Incomplete *mrkABCDF* and *fimACDH* operons were found in 18 (14.63%) and 11 (8.94%) isolates, respectively. None of the isolates carried complete *mrk* and *fim*

operons simultaneously (Table S2). We randomly selected 13 complete *mrk* and 7 complete *fim* positive strains, and transconjugants harboring *fim* and *mrk* operons were successfully obtained from 5 of 7 (71.43%) and 2 of 13 (15.38%) isolates, respectively. All 7 transconjugants carrying *fim* or *mrk* operons were strong biofilm producers in comparison with the recipient strains *E. coli* C600 (Fig. S1). In addition, all the five transconjugants carrying *fim* operons showed resistance to ampicillin, florfenicol, sulfamethoxazole/trimethoprim and olaquindox, and three ones were resistant to gentamycin (Table S2). The resistant phenotype of florfenicol, colistin, tetracycline, and sulfamethoxazole/trimethoprim were co-transferred with *mrk* operons among two transconjugants carrying *mrk* operons, and one also reduced susceptibility to ampicillin and ceftiofur, and the other one was resistant to gentamycin.

Furthermore, we explore the distribution of *mrkABCDF* and *fimACDH* operons among 6 Enterobacteriaceae using archived WGS data. The prevalence of *mrkABCDF* was highest in *K. pneumoniae* (100%, 9457/9457) followed by *Enterobacter cloacae* (6.54%, 14/214), *C. freundii* (3.94%, 13/330), and *E. coli* (0.62%, 199/21387). In contrast, *fimACDH* was present in *E. cloacae* (62.15%, 133/214) followed by *Citrobacter freundii* (0.30%, 1/330), *E. coli* (0.28%, 60/21387), and *K. pneumoniae* (0.08%, 8/9457). Neither *mrkABCDF* nor *fimACDH* was found among *Salmonella* spp. ($n = 12{,}535$) or *Proteus mirabilis* ($n = 265$) (Table S3).

**Characterization of plasmids encoding types 1 and 3 fimbriae and phylogenetic analyses.** *E. coli* strains 2010FS332 carrying *mrkABCDF* and strain 2005FS026 carrying *fimACDH* were selected for further sequencing using the Nanopore sequencing platform. The combined MiSeq and Nanopore sequencing data yielded the complete sequence of the endogenous plasmids from these strains; p2010FS332 (accession no. OK217279) and p2005FS026 (accession no. OK236218), respectively.

The *mrk*-positive plasmid p2010FS332 was an IncX1 type (48.958 kb), and its backbone sequence was almost identical to an *mrk*-bearing IncX1 plasmid from the *E. coli* isolate pMAS2027 (accession no. FJ666132). The basic core structure of the IncX plasmid group was shared and included *pir-bis-par-hns-topB-taxB-pilX-actX-taxCA*. The primary differences between these plasmids were the region located between resolvase and *hns* where *mrkABCDF* was embedded and bracketed by IS*903* and *insA/B* (Fig. 4a). The remaining 7/12 *E. coli* isolates harboring *mrkABCDF* and IncX1 replicons were highly similar to p2010FS332 and to 8 *mrk*-IncX1 plasmids archived in GenBank (Fig. 4b).

We further explored the distribution and evolution of *mrkABCDF* using 50 *mrkABCDF* operons from the GenBank data archive. This group contained 7 chromosomally-encoded *mrk* operons that were represented in numerous species including *K. pneumoniae*, *K. aerogenes*, *E. hormaechei*, *E. coli*, *C. freundii*, and *C. koseri*. The remaining 43 strains contained plasmid-encoded *mrk* gene clusters that were present on numerous plasmid replicon types including IncX1, IncFIB, IncFIA, IncFII, IncHI1, IncHI2, IncA/C, and IncR. Strains possessing *mrkABCDF* operon had diverse origins and it included humans, food animals (pig, cattle, chicken, duck, goose), food and environment (water), with water being the most predominant one ($n = 18$, 36%), followed by humans ($n = 15$, 30%) and pigs ($n = 8$, 16%) (Fig. 4c and Fig. S2a). The sampling locations were also from 13 countries, and the most predominant one was UK ($n = 23$, 46%), followed by China ($n = 6$, 12%) (Fig. 4c and Fig. S2b).

The *fimACDH*-carrying plasmid p2005FS026 that we completely sequenced (150.191 kb) contained a typical IncFII replication region (2.7 kb), which comprises *repA1*, *repA4*, *repA3*, and *repA2*. This plasmid was almost identical to *fim*-carrying IncFII plasmid p253 (accession no. MT648288) from a pig *E. coli* isolate reported in our previous study (11) except for an inversion of the multidrug resistance region (MRR) (~33.5 kb), containing 11 ARGs including *oqxAB* and *floR* (Fig. 5a). The *silABCESP* (~12.5 kb) and *fimACDH* were upstream and downstream of the MRR, respectively, and the former were bracketed by two copies of *insA/B* while the latter were flanked by IS*Ec63* and *insA/B*. Furthermore, the p2005FS026 and p253 backbones were highly similar to a *fim*-carrying IncFII plasmid pE11019 that completely lacked ARG sequences (accession no. CP035752). The remaining single *E. coli* isolate from this group that harbored the *fim* operon and IncFII replicons in our study was almost identical to the *mrk* plasmids p2005FS026 and p253 (Fig. 5b).

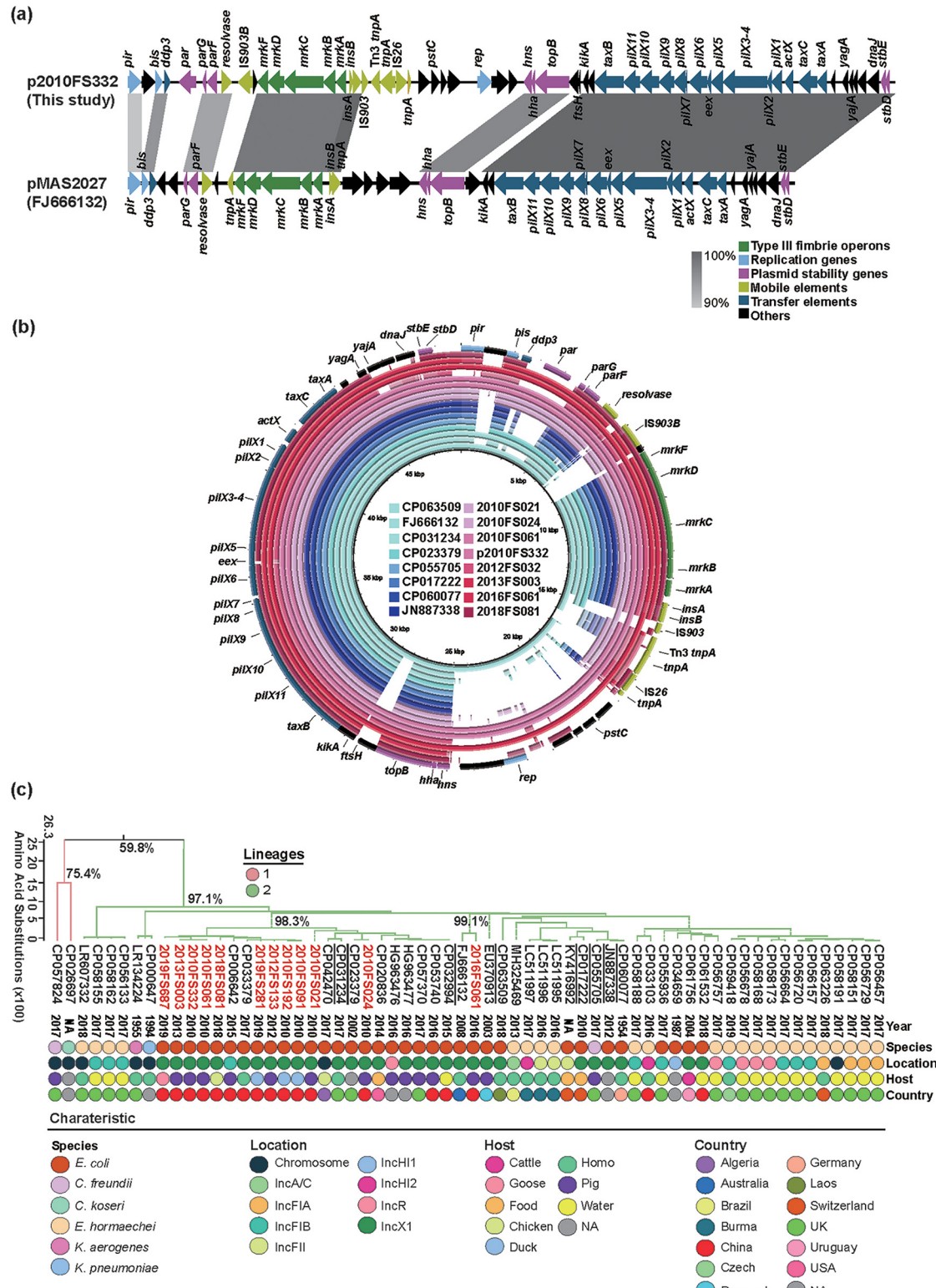

**FIG 4** Characteristics of the plasmids encoding *mrkABCDF* and phylogenetic analysis. (a) Linear sequence alignments of *mrkABCDF*-carrying plasmids p2010FS332 (this study) and pMAS2027 (accession no. FJ666132). (b) Circular sequence alignment of *mrkABCDF*-carrying plasmids in this study and other similar plasmids from the GenBank database as indicated. (c) Phylogenetic analysis between the concatenated *mrk* operons in this study and those from GenBank were conducted by building Maximum likelihood (ML) trees based on amino acid sequences.

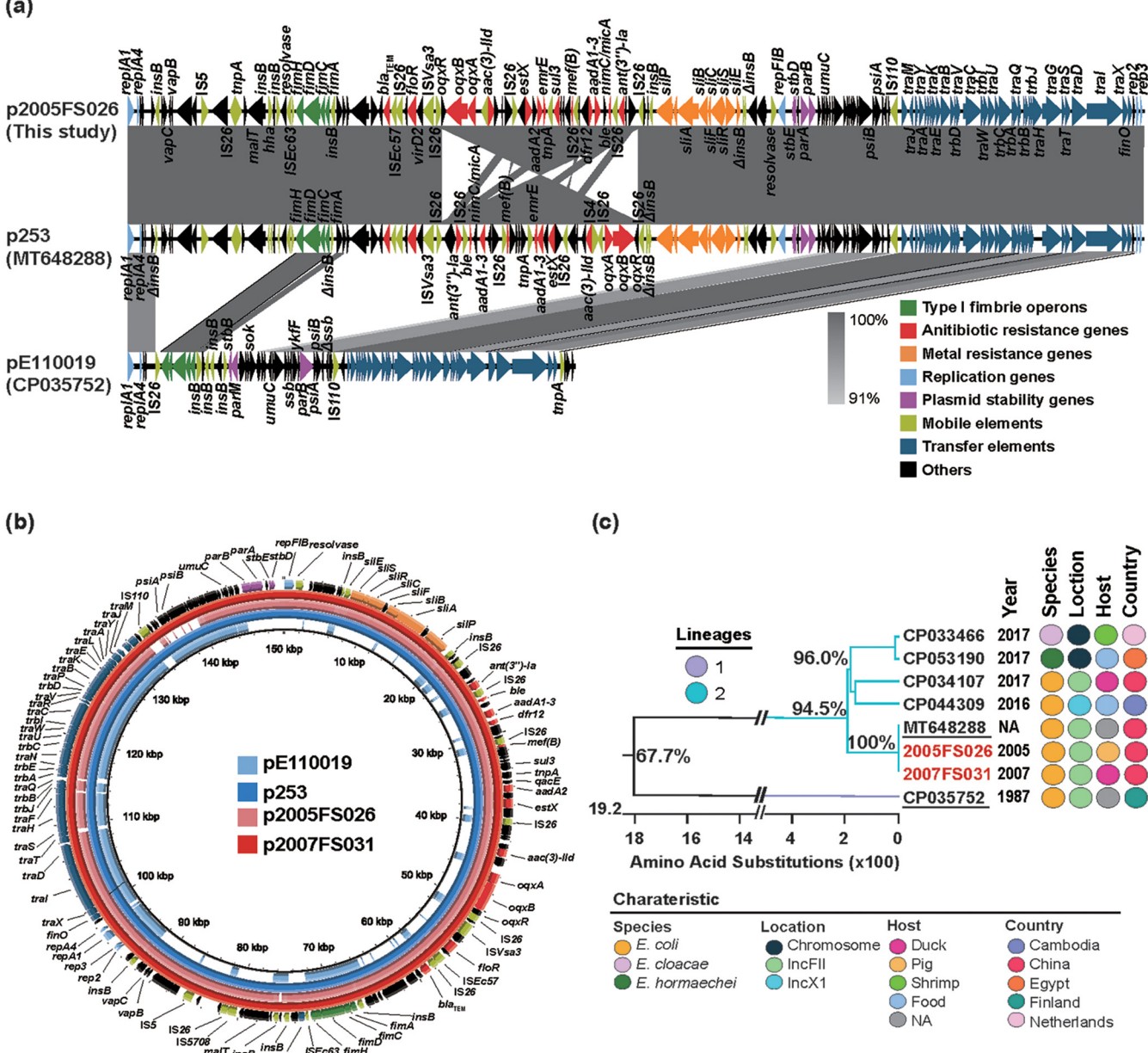

**FIG 5** Characteristics of plasmids encoding *fimACDH* and phylogenetic analysis. (a) Linear sequence alignment of *fimACDH*-carrying plasmids p2005FS026 (this study) and p253 and pE110019 (as indicated). (b) Circular sequence alignment of *fimACDH*-carrying plasmids in this study and other similar plasmids from the GenBank database as indicated. (c) Phylogenetic analysis between the concatenated *fim* operons in this study and those from GenBank were conducted by building Maximum likelihood (ML) trees based on amino acid sequences.

We compared our data with the 6 *fimACDH* operons available in the GenBank data archive, and these 6 included 2 that were located on chromosomes (*E. cloacae* and *E. hormaechei*) and 4 present on plasmids (3 IncFII and 1 IncX1 replicons). These 6 *fimACDH* strains were obtained from ducks, shrimp, and food from 5 countries. A phylogenetic reconstruction indicated that the 2 *fim* operons found in our study and 5 of the 6 archived sequences clustered together with an amino acid sequence identity of ≥94.5% (Fig. 5c).

## DISCUSSION

The current study investigated the biofilm forming ability and the potential molecular mechanism among Enterobacteriaceae isolates from a large collection of diseased food producing animals. We found that 7.01% of the isolates were strong biofilm producing *E. coli*

and the prevalence of this phenotype was less than that found for uropathogenic *E. coli* (UPEC) isolates (24.8%) (19). The plasmid-mediated type 3 fimbriae encoded by the *mrkABCDF* operon and type 1 fimbriae encoded by *fimACDH* operon were identified to confer a strong biofilm-forming phenotype to laboratory strains of *E. coli* as previously described (10, 11). The complete *mrkABCDF* and *fimACDH* clusters were present in 34.96% and 5.7% of all strong biofilm producing *E. coli*, respectively, indicating that these two operons are major biofilm-associated factors in *E. coli*.

In the present study, most of the strong biofilm producers were also MDR strains. In particular, they possessed significantly higher resistance levels to olaquindox and florfenicol when compared with non-strong biofilm producers. Consistently, WGS analysis showed that the strong biofilm producers harbored multiple ARGs including the florfenicol resistance gene *floR* and the MDR efflux pump *oqxAB* conferring resistance to olaquindox and ciprofloxacin. Furthermore, *oqxAB* and *floR* were identified to colocate with *mrkABCDF* or *fimACDH* on conjugative IncX1 and IncFII plasmids in the present and previous studies (10, 11, 17, 20). Olaquindox has been used as a growth promoter in pigs until May 2018 in China, and florfenicol was also used in Chinese pig farms especially for the treatment of swine respiratory disease. The close genomic proximity of *oqxAB* and *floR* to biofilm-encoding genes indicated that these traits could be coselected under florfenicol and olaquindox selective pressure. This might partially explain the reason why *E. coli* isolates of pig origin had more strong biofilm producers than from other origins in the present study. The cospread of ARGs and biofilm-associated factors on conjugative plasmids is especially worrisome because this could compromise the already few treatment options available for infections caused by plasmid-encoded type 1 and 3 fimbria-producing MDR *E. coli*.

The *mrk* gene clusters have been associated with a wide range of conjugative plasmid types including IncX1, IncFIA, and IncFIB (10, 18). In the current study, the *mrkABCDF* operons (61.5%) were mostly located on IncX1 plasmids with highly similar backbones and were also highly similar to *mrk*-carrying IncX1 plasmids from 3 members of Enterobacteriaceae of diverse origins including humans, food animals, food, and pets from 6 different countries. The presence of *mrkABCDF* was found to enhance conjugation frequencies by promoting biofilm formation, and this might facilitate the spread of *mrk*-positive plasmids among the Enterobacteriaceae (10, 21).

These results indicated that *mrkABCDF* has been mobilized to diverse species of Enterobacteriaceae *via* mobile genetic elements and in particular, conjugative IncX1 plasmids. Indeed, the MrkABCDF displayed high levels of amino acid identities to their counterparts on plasmids of differing replicon types and on chromosomes of diverse Enterobacteriaceae species that originated from food animals, humans, food and the environment across the globe, in particular UK and China. Furthermore, *mrkABCDF* was prevalent in 4/6 Enterobacteriaceae species especially *K. pneumoniae* (100%) from the GenBank data archive NCBI. Taken together, these results were consistent with the most likely origin for the *mrk* operon, *K. pneumoniae*, which has then become widespread among the Enterobacteriaceae, including *E. coli* (10, 18).

Both type 1 and 3 fimbriae belong to the group of chaperone-ushered pili that are assembled at the outer membrane by a periplasmic chaperone and an usher protein (17, 22, 23). The plasmid-encoded type 3 fimbriae are comprised of the major (MrkA) and minor (MrkF) subunits, as well as chaperone (MrkB), usher (MrkC), and adhesin proteins (MrkD) (10, 22). The biofilm formation was deficient when the insertion was localized to the mrk operon, specifically in the *mrkA*, *mrkC* or *mrkD* genes (10, 16). By contrast, the plasmid-encoded Type 1 fimbriae are encoded by 4 contiguous genes (*fimACDH*) where FimC and FimD play the roles of putative chaperone and usher, respectively, however, roles for FimA and FimH have not been identified. The biofilm formation in E. coli was determined by an intact *fimACDH* gene operon in our previous study (11). Unlike plasmid-mediated *fimACDH*, the chromosomally-encoded type 1 fimbriae is frequently present in the majority of Enterobacteriaceae (14, 18, 24), and significant heterogeneity exists between DNA sequences encoding type 1 fimbriae among Enterobacteriaceae (25, 26). The biosynthesis and structure of chromosomal type 1 fimbriae in *E. coli* and *S.* Typhimurium have been

extensively studied, and it encodes a major subunit FimA and a minor tip adhesin FimH (14, 25, 26). A future goal is to explore the structures and function of the type 1 fimbrial encoded by the *fimACDH* operon, and the roles of each of FimACDH in biofilm formation.

In conclusion, the plasmid-encoded type 3 fimbriae encoded by *mrkABCDF* and type 1 fimbriae encoded by *fimACDH* were major contributors to enhancing biofilm formation among strong biofilm *E. coli* from diseased food producing animals. These two operons were found to coexist with ARGs on conjugatable IncX1/IncFII plasmids with similar backbones, respectively. Our findings also revealed that these two operons, in particular *mrk*, were present both on plasmids of various replicon types and on chromosomes from diverse Enterobacteriaceae species from numerous origins and countries, indicating a potential antibiofilm target. This is the first comprehensive report of the prevalence, evolution, and transmission of plasmid-encoded type 1 and 3 fimbriae among the Enterobacteriaceae. Future studies are necessary to investigate the transmission and function of these two operons to better understand their potential threats to public health and for the screening of small-molecule biofilm inhibitors.

## MATERIALS AND METHODS

**Bacterial strains, biofilm formation assay, and antimicrobial susceptibility testing.** We isolated 1,753 non-duplicate Enterobacteriaceae strains from diseased food-producing animals that included 1,272 avian samples (888 duck, 218 chicken, and 166 goose) and 481 samples from pigs from >100 farms throughout Guangdong province, China, from 2002 to 2019. These isolates were recovered directly from fecal samples or swabs from animal organs (liver, heart, or lung) from farms or diagnostic laboratories as previously described (27).

Quantification of static biofilm production was performed using 96-well flat-bottom polystyrene microtiter plates using crystal violet, and the extent of biofilm formation was determined as previously described (28). All strong biofilm producers were identified by matrix-assisted laser desorption/ionization-time-of-flight mass spectrometry and 16S rRNA gene sequence-based analyses. Antimicrobial susceptibilities of the tested isolates were determined using the agar dilution method, and the results were interpreted according to the Clinical and Laboratory Standards Institute (CLSI, 2018: M100-S28) (29), veterinary CLSI (VET01-A4E/VET01-S3E) (30) (supplemental Materials and Methods).

**Genetic characterization of strongly biofilm-forming isolates.** Total genomic DNA of 42 strong biofilm-forming *E. coli* isolates were extracted by using the TIANamp Bacteria DNA Kit (Tiangen, China). The quality and concentration of the bacterial genomic DNA were evaluated via electrophoresis on a 1% agarose gel and analysis on a NanoDrop2000 system (Thermo Scientific, Waltham, MA, USA) and a Qubit 3 Fluorometer (Thermo Scientific, Waltham, USA). Illumina libraries were prepared and sequenced using Illumina HiSeq 4000 platform (San Diego, CA, USA) as 150-bp paired-end reads. Adaptors and low-quality bases were trimmed with Trimmomatic v0.38 (31), and reads qualities were assessed using FastQC v0.11.6 (https://www.bioinformatics.babraham.ac.uk/projects/fastqc/) and MultiQC v1.7 (32). High-quality reads were *de novo* assembled with SPAdes v3.6.2 to generate genome contigs (33). Sequencing quality and statistics per isolates were checked using the QualiMap v2.2.2 (34). Genome assemblies' quality was assessed with QUAST v5.0.2 (35) and contigs of less than 200 bp were filtered out. Examinations of known plasmid replicon, and antibiotic resistance genes were carried out using ABRicate v1.0.1 (https://github.com/tseemann/abricate) (>80% identity and >80% coverage). Reference sequences of plasmid and antibiotic resistance genes were from databases PlasmidFinder (36), and ResFinder (37), respectively. Multilocus sequence typing (MLST) were performed by MLST v2.19.0 (https://github.com/tseemann/mlst). In addition, *in silico* phylotyping of *E. coli* was carried out using the Clermon Typing method (38). Further, assemblies from all isolates were mapped to the reference sequence 2013FS003 using Snippy v4.6.0 (https://github.com/tseemann/snippy). Single nucleotide polymorphisms (SNPs) were called and recombinant regions were removed using Gubbins v2.4.1 (39). RAxML v8.2.12 (GTRGAMMA substitution model) with 100 bootstrap replicates to assess support was used to construct a phylogenic tree and visualized with iTOL v4 (40). The population structure of each phylogenetic tree was defined using hier-BAPS v6.0 (41).

**Characterization of plasmids carrying *fim*/*mrk* operons.** The transferability of type 1/3 fimbriae operons was examined using conjugation experiments using the streptomycin-resistant *E. coli* C600 as the recipient. All transconjugants were tested for antimicrobial susceptibility and biofilm formation as described above. To obtain the complete sequence of plasmids encoding type 1/3 fimbriae, two isolates (2010FS332 and 2005FS026) encoding type 1 and 3 fimbriae, respectively, were selected for long-read sequencing using ONT Gridion Platform (Nanopore, Oxford, UK) (42). DNA extraction and quality control were performed as previously described. An Oxford Nanopore MinION 9.4.1 flowcell and SQK-RBK004 rapid sequencing kit was used with base calling by Guppy v3.1.5 (https://nanoporetech.com/nanopore-sequencing-data-analysis). *De novo* hybrid assembly using both short reads (Illumina) and long reads (ONT) was performed using Unicycler v0.4.4 (43). The Contigs were further polished with Pilon v1.23 (44) through three iterations. Gene prediction and annotated were performed by the RAST tool (45) (https://rast.nmpdr.org/), ISFinder (46), BLAST (https://blast.ncbi.nlm.nih.gov/Blast.cgi) and rechecked manually. The sequence comparison and map generation of plasmids encoding type 1/3 fimbriae were performed using Easyfig (47) and the BLAST Ring Image Generator (48). The complete sequence of plasmid p2010FS332 and p2005FS026 has been deposited in GenBank under accession no. OK217279 and accession no. OK236218, respectively.

**Function, prevalence, and phylogeny of type 1/3 fimbriae operon genes.** The *mrk* and *fim* operons from the strong biofilm-forming *E. coli* strains 2010FS332 and 2005FS026, respectively, were amplified,

and PCR amplicons were then ligated to expression vector pMD19-T and introduced into *E. coli* strain DH5$\alpha$ by chemical transformation (Table S1). Colonies were selected on LB agar supplemented with ampicillin (100 mg/mL). Transformants were randomly screened for the presence of the *mrk* and *fim* operons by PCR assay and sequencing, and the resultant clones were tested for biofilm formation as described above. Biofilm structures were further examined using SEM (supplemental Materials and Methods).

To understand the spread of the *mrk/fim* operons, all *E. coli* that were strong biofilm producers were screened for the presence of *mrk* and *fim* genes using PCR (Table S1). Additionally, *mrk* and *fim* operons were *in silico* identified from Enterobacteriaceae members that possessed WGS data from a public data-base (https://www.ncbi.nlm.nih.gov/datasets). Phylogenetic correlations were constructed between con-catenated *mrk* and *fim* operons from this study and those from GenBank using maximum likelihood (ML) trees based on predicted amino acid sequences (supplemental Materials and Methods).

**Statistical analysis.** Statistical significance for comparison of prevalence data and proportions was performed in R using the $\chi^2$ test. Other data were statistically analyzed using GraphPad Prism v8.0.1 software. *P* values of $<$0.05 were deemed to be statistically significant. Specific tests of statistical significance are detailed in figure legends and table footnotes.

**Data availability.** All genome assemblies of these strains were deposited in GenBank and were reg-istered under BioProject accession number PRJNA747154.

## SUPPLEMENTAL MATERIAL

Supplemental material is available online only.

**SUPPLEMENTAL FILE 1**, PDF file, 0.3 MB.
**SUPPLEMENTAL FILE 2**, XLSX file, 0.03 MB.

## ACKNOWLEDGMENTS

This work was supported by the Guangdong Major Project of Basic and Applied Basic Research, No. 2020B030103007, National Natural Science Foundation of China (31802244), Local Innovative and Research Teams Project of Guangdong Pearl River Talents Program (2019BT02N054), Program for Changjiang Scholars and Innovative Research Team in University of Ministry of Education of China (IRT_17R39), and Innovation Team Project of Guangdong University (2019KCXTD001).

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
