## [Reviewer comments · Microbiology Spectrum]

Microbiology Spectrum

Molecular Epidemiology of Plasmid-Mediated Types 1 and 3 Fimbriae Associated with Biofilm Formation in Multidrug Resistant *Escherichia coli* from Diseased Food Animals in Guangdong, China

Wen-Ying Guo, Hui Zhang, Ming Cheng, Min-Rui Huang, Qian Li, Yu-Wei Jiang, Ji-Xing Zhang, Ruan-Yang Sun, Min-Ge Wang, Xiao-Ping Liao, Ya-Hong Liu, Jian Sun, and Liang-Xing Fang

Corresponding Author(s): Liang-Xing Fang, South China Agricultural University

Review Timeline:

Submission Date:	December 4, 2021
Editorial Decision:	April 19, 2022
Revision Received:	May 9, 2022
Accepted:	June 6, 2022

Editor: Ilana Kolodkin-Gal

Reviewer(s): The reviewers have opted to remain anonymous.

Transaction Report:

DOI: <https://doi.org/10.1128/spectrum.02503-21>

April 19, 2022

Dr. Liang-Xing Fang
South China Agricultural University
Guangzhou, Guangdong
China

Re: Spectrum02503-21 (Molecular Epidemiology of Plasmid-Mediated Types 1 and 3 Fimbriae Associated with Biofilm Formation in Multidrug Resistant *Escherichia coli* from Diseased Food Animals in Guangdong, China)

Dear Dr. Liang-Xing Fang:

Link Not Available

Sincerely,

Ilana Kolodkin-Gal

Journals Department
Reviewer comments:

Reviewer #1 (Comments for the Author):

The work done represent a nice piece of work by the authors however there are few basic queries that needs to be clarified before going on taking any decision for publication:

1. The major goal of study is not clear. For instance the title depicts the work is on "Plasmid-Mediated Types 1 and 3 Fimbriae 2 Associated with Biofilm Formation in Multidrug Resistant *Escherichia coli*" however there are already many studies on such comparison. The clear contrast in comparison to other studies has not been given in abstract.
2. The sequencing details of genomes (42 strong biofilm-forming *E. coli*) are not given. There should be a detailed paragraph on the library size, assembles and commands for both the sequencing methods along with the details of tool used for basic annotations.

3. Discussion part is over all placed and not in a perfect story line. For examples Fimbriae explained in one paragraph followed by plasmid than suddenly the biofilm details. The multiple breaks do not highlight the clear outcome and thus diluting the impact over all. This can be improved by revising the discission part.
4. There is no major comment on the association of E.coli with specific organ types, geographical locations etc. which has been displayed in images but not in manuscript.
5. It is suggested to also perform Protein-Protein Interaction network for the identified plasmids with core genes to predict occurrences of gene recombination and HGT events.

Staff Comments:

Preparing Revision Guidelines

Please return the manuscript within 60 days; if you cannot complete the modification within this time period, please contact me. If you do not wish to modify the manuscript and prefer to submit it to another journal, please notify me of your decision immediately so that the manuscript may be formally withdrawn from consideration by Microbiology Spectrum.

Replies to Reviewer 1:

1. The major goal of study is not clear. For instance the title depicts the work is on "Plasmid-Mediated Types 1 and 3 Fimbriae Associated with Biofilm Formation in Multidrug Resistant *Escherichia coli*" however there are already many studies on such comparison. The clear contrast in comparison to other studies has not been given in abstract.

Answers: We agree with the criticism that the major goal of our study is not clear in our first manuscript. Types 1 and 3 fimbriae in Enterobacteriaceae play versatile roles in bacterial physiology including attachment, invasion, cell motility as well as with biofilm formation and urinary tract infections. Interestingly, Types 1 and 3 fimbriae are not only encoded on chromosomes but also conjugative plasmids. Indeed, there are already some studies on the mobile types 1 and 3 fimbriae involved in biofilm formation among Enterobacteriaceae, however, the prevalence, transmission and homology of the mobile types 1 and 3 fimbriae among Enterobacteriaceae is still largely unknown. Furthermore, it also remains unknown whether the mobile types 1 and 3 fimbriae could be used as potential anti-biofilm target. In our study, we investigated the biofilm forming abilities and drug susceptibilities in a large collection of Enterobacteriaceae isolates from diseased food producing animals. We further explored the prevalence, function, evolution and transmission of plasmid-encoded type 1/3 fimbriae. We have the information in the abstract and renarrated it as you suggested.

2. The sequencing details of genomes (42 strong biofilm-forming *E. coli*) are not given. There should be a detailed paragraph on the library size, assembles and commands for both the sequencing methods along with the details of tool used for basic annotations.

Answers: Thanks for your suggestion. We have added the information in supplementary data, and in the main text as you suggested. Please see lines 127-153, and 161-178 in the revised manuscript, and the supplementary data (Whole genome sequencing and assembly information).

3. Discussion part is over all placed and not in a perfect story line. For examples Fimbriae explained in one paragraph followed by plasmid than suddenly the biofilm details. The multiple breaks do not highlight the clear outcome and thus diluting the impact over all. This can be improved by revising the discussion part.

Answers: We agree with the criticism and have modified the discussion part as suggested. Please see the discussion in the revised manuscript.

4. There is no major comment on the association of *E. coli* with specific organ

types, geographical locations etc. which has been displayed in images but not in manuscript.

Answers: Thanks for your suggestion. We have added the information and please see lines 311-317 and 383-387 in the revised manuscript.

5. It is suggested to also perform Protein-Protein Interaction network for the identified plasmids with core genes to predict occurrences of gene recombination and HGT events.

Answers: Thanks for your suggestion. It's interesting to perform Protein-Protein Interaction network for the identified plasmids with core genes to predict occurrences of gene recombination and HGT events, but we are sorry that we don't quite understand the purpose of it, and it's also a little difficult for us to perform it. Furthermore, in our study, we have performed conjugation experiments to determine the transferability of plasmid bearing type 1/3 fimbriae operons. What's the difference between performing Protein-Protein Interaction network and conjugation experiments? If we should do Protein-Protein Interaction network, could you give us some guidance? Thanks!

June 6, 2022

Dr. Liang-Xing Fang
South China Agricultural University
Guangzhou, Guangdong
China

Re: Spectrum02503-21R1 (Molecular Epidemiology of Plasmid-Mediated Types 1 and 3 Fimbriae Associated with Biofilm Formation in Multidrug Resistant *Escherichia coli* from Diseased Food Animals in Guangdong, China)

Dear Dr. Liang-Xing Fang:

While I could only secure one reviewer, I read the paper myself and I believe that it is of interest and solid. I recommend accepting this paper for publication

Your manuscript has been accepted, and I am forwarding it to the ASM Journals Department for publication. You will be notified when your proofs are ready to be viewed.

Sincerely,

Ilana Kolodkin-Gal
Editor, Microbiology Spectrum

Journals Department
Supplemental Dataset: Accept
Supplemental Material: Accept